# Re-analysing Ebola spread in Sierra Leone: The importance of local social dynamics

**Paul Richards[1]\*, Gelejimah Alfred Mokuwa[2], Ahmed Vandi[3], Susannah Harding Mayhew[4], Ebola Gbalo Research Team[¶]**

1 School of Environmental Sciences, Njala University, Mokonde, Sierra Leone, 2 Department of Agriculture and Forestry, Eastern Polytechnic, Kenema, Sierra Leone, 3 School of Community Health Sciences, Kowama, Bo, Sierra Leone, 4 Department of Global Health and Development, London School of Hygiene and Tropical Medicine, London, United Kingdom

¶ Membership of the Ebola Gbalo Research Team is provided in the Acknowledgments.
\* paul.richards1945@gmail.com

**Data Availability Statement:** The data are available in the LSHTM data archive with the following DOI: https://doi.org/10.17037/DATA.00001830

## Abstract

### Background

The 2013–15 Ebola epidemic in West Africa was the largest so far recorded, and mainly affected three adjacent countries, Guinea, Liberia and Sierra Leone. The worst affected country (in terms of confirmed cases) was Sierra Leone. The present paper looks at the epidemic in Sierra Leone. The epidemic in this country was a concatenation of local outbreaks. These local outbreaks are not well characterized through analysis using standard numerical techniques. In part, this reflects difficulties in record collection at the height of the epidemic. This paper offers a different approach, based on application of field-based techniques of social investigation that provide a richer understanding of the epidemic.

### Methods

In a post-epidemic study (2016–18) of two districts (Bo and Moyamba) we use ethnographic data to reconstruct local infection pathways from evidence provided by affected communities, cross-referenced to records of the epidemic retained by the National Ebola Response Commission, now lodged in the Ebola Museum and Archive at Njala University. Our study documents and discusses local social and contextual factors largely missing from previously published studies.

### Results

Our major finding is that the epidemic in Sierra Leone was a series of local outbreaks, some of which were better contained than others. In those that were not well contained, a number of contingent factors helps explain loss of control. Several numerical studies have drawn attention to the importance of local heterogeneities in the Sierra Leone Ebola epidemic. Our qualitative study throws specific light on a number of elements that explain these heterogeneities: the role of externalities, health system deficiencies, cultural considerations and local coping capacities.

**Funding:** MRC Grant: MR/N015754/1 SHM, UK Medical Research Council Grant: MR/N015754/1, PR, UK Medical Research Council Grant: MR/N015754/1 Grant title: Building resilient health systems: lessons from international, national and local emergency responses to the Ebola epidemic in Sierra Leone. The funder played no role in study design, data collection and analysis, decision to publish or preparation of the manuscript.

**Competing interests:** The authors have declared that no competing interests exist.

## Conclusions

Social issues and local contingencies explain the spread of Ebola in Sierra Leone and are key to understanding heterogeneities in epidemiological data. Integrating ethnographic research into epidemic-response is critical to properly understand the patterns of spread and the opportunities to intervene. This conclusion has significant implications for future interdisciplinary research and interpretation of standard numerical data, and consequently for control of epidemic outbreaks.

## Introduction

Ebola Virus Disease (EVD) is a viral hemorrhagic fever largely spread through contact with body fluids of an infected person. From its first identification from an outbreak close to the Ebola river in the Democratic Republic of Congo in the 1970s there have been over 20 outbreaks, mainly in isolated communities in the central African forest belt. In 2013 there was an outbreak in south-eastern Guinea, at the western extremity of the humid forested region in West Africa. Infection rapidly spread to the neighboring countries of Liberia and Sierra Leone, largely through inter-personal contact. Carers–whether medical personnel or family members–are especially vulnerable to infection, through nursing the sick or preparing bodies for burial. The disease became established in urban centers, and cases spread to Europe and North America. A major international effort was mounted to contain the West African outbreak. By the end of 2015 this effort was largely successful, though the virus has resurfaced since in two parts of the Democratic Republic of Congo. To date, however, the West African outbreak remains the largest on record with 28 616 confirmed, probable and suspected cases and 11 310 deaths by June 2016 [1].

As the first large-scale regional outbreak the West African episode has attracted considerable analytic attention. There is recognition that lessons needed to be learned relevant to other potential large-scale episodes, such as the one that occurred in North Kivu (DRC) [2]. Some of the key epidemiological literature on the West African outbreak deploys what we will here term the standard numerical approach (for example [1, 3, 4]). This makes considerable use of multivariate statistical methods and models. Applied to the West African epidemic of EVD these methods lead to the identification of numerous local-level heterogeneities. Numerical methods are only as reliable as the data on which they feed and collecting reliable data in the heat of an emergency response to an Ebola epidemic is far from straightforward. Whether heterogeneities apparent in numerical data arise from problems of data collection or complexities on the ground is unclear. Our own work, based on fieldwork in two districts in Sierra Leone, the worst affected country in terms of confirmed cases of Ebola, is intended to address this interpretive challenge. Our research strategy seeks to reconstruct local infection chains through evidence provided by communities, using ethnographic and micro-geographical data gathering techniques. Ebola spread from person to person, meaning that it has an inherently social dimension. This social dimension, we will argue, is key, to understanding heterogeneities in the epidemiological data. Our aim, therefore, is to understand the spread of Ebola in two largely rural districts of central Sierra Leone by assessing the social dynamics of infection and the implications of this for control of epidemic outbreaks in many other settings.

### Limitations of previous numerical studies

We examined papers using the standard numerical approach to explore data from standard reporting forms and laboratory records [1, 3–6]. These papers exhibit awareness of possible defects in the data set, and exercise care in reaching conclusions. Garske et al. [1] remark that "*unfortunately, no information on the mechanisms by which individual cases enter the database was recorded, and it is therefore not possible to assess the impact this has on [case fatality] estimates*" (p. 7). After applying editing rules such as the elimination of incomplete records, they reach a conclusion that heterogeneities are real and not just artefacts of noisy data. A picture is presented of Ebola in West Africa not as a single epidemic but as a concatenation of local and at times dissimilar events. We will provide further (qualitative) evidence for supposing that this conclusion is sound.

The cited papers establish between them a number of broad, descriptive findings about Ebola on an epidemic scale–for example, that risks of cross-infection ran in households and peaked during the final course of the disease and subsequent burials, that the disease affected men and women equally, and that infants and the elderly were especially vulnerable. But on questions with especial significance to protection, such as whether infection was due to local nursing and burial customs or more prosaic practices such as cleaning beds and corpses, or whether the susceptibility of the very young and the elderly was by reason of age, rather than because of body contact between infected mother and child or the exercise of the special responsibilities of the elderly to the sick and dying, the numerical data do not speak.

It would be good to know whether local disputes, variations in ritual practices, or bad luck modified infection risks. This ambition is clearly articulated in the paper by Fang et al. [3]. They surmise that "*the difference in incidence rates among ethnic groups might be due to their geographic locations, economic development, social behaviors, or religious traditions*" and propose that "*further investigations are needed to elucidate this issue*" (p. 4492).

Our own paper takes this call as its starting point. Adopting an anthropological approach, we visited a majority of nodes in the main infection chains identified by eye-witnesses and survivors in two districts, Bo and Moyamba. We then traced with these informants how the various infection chains unfolded. We identified chiefdoms and villages that had suffered contrasting outbreaks (large, small, quickly contained, long-lasting outbreaks) and for each we identified the probable index cases then sought to trace the stories of how the infection spread from these cases to infect others. This information was then cross-checked with the Njala data base records, and with a document kept by a nurse-volunteer at the Moyamba Ebola holding center, that records in careful detail the admission of patients and their test results in the first phase of the epidemic. This proved helpful in confirming statements made by eyewitnesses but not covered in the national Ebola data base. The result of linking quantitative and quantitative information was a richer account of local heterogeneities in infection patterns and response that provides the basis for a wider understanding of the issues that explain heterogeneities in epidemics.

## Methods

The findings reported in this paper are part of a larger study (*Ebola Gbalo*–Ebola Trouble) analyzing different levels of response–community, district, national and international–to the Ebola crisis in Sierra Leone 2014–15, and the interactions between these various levels [2]. An aim of this larger project was to assess national and local capacities for response to the epidemic, and to place these elements, often masked by the more highly visible international efforts, into a broader collaborative context. Data were collected principally by the second and third authors at various time during 2016 and 2017. Data on the infection chain in Niawa Lenga chiefdom were collected in 2018 by a team of three people led by the first author (see acknowledgements).

To understand epidemic responses, we first needed a more detailed picture of how infection chains were initiated and sustained, and how they were ended more rapidly in some cases than in others. The purpose of the work described in this paper was to establish what combinations of local factors determined the heterogenous responses to Ebola already determined to have occurred via application of the standard numerical approach. We used ethnographic techniques to elucidate the behavioral factors behind the numbers.

Ethics approval was received from the ethics review boards of the London School of Hygiene and Tropical Medicine (Approval Reference Number 12016) and Njala University (Institutional Review Board 2016).

## Sample strategy and ethnographic fieldwork approach

While not possible to apply an ethnographic approach on a national scale, even in a small country like Sierra Leone, it is nevertheless important to examine how local infection chains were concatenated. How did infection chains start off, and die down, and how were adjacent chains related? This suggested a geographically focused case study might be better than a spread of randomly chosen snapshots from different parts of the country.

A further criterion was to choose a time period after the initial stages (when the epidemic was concentrated in the east of the country), but before the international response was fully ramped up, so that the functioning of local agency would be more readily visible. These criteria led to the selection of two adjacent districts–Bo and Moyamba–in the southern part of the country.

The propensity for EVD to move along main roads in Sierra Leone, emphasized in previous studies [3, 7, 8], provided the basis for our sampling strategy. We followed infection chains down the main roads crossing Bo and Moyamba Districts in the direction of Freetown, while also taking careful account of side branches connecting interior villages, where infection was blocked or rapidly terminated due to the propensity of rural communities to go into "lock down" mode when threatened by dangers. We describe key interconnected infection chains from both districts.

Members of our research team are based at Njala University, an institution with campuses at Mokonde in Kori Chiefdom, in Moyamba District, and in Bo. Our group had earlier undertaken fieldwork in Ebola affected communities for the Ebola Response Anthropology Platform (www.ebola-anthropology.net) in 2014–15, and so knew key localities where enquiries were needed. We began with the earliest cases and traced connections from these known nodes. A number of villages were identified for further investigation. We checked these choices against the numbers of confirmed Ebola cases per chiefdom, and then by village, in the national Ebola data base, a copy of which is archived in Njala, and as a result added some places to the sample of communities where enquiries were required.

Local authorities were then notified. Community-level informed consent for our enquiries was obtained in meetings held to explain the project attended by local chiefs and elders. Key informants were then identified (e.g. survivors, or members of affected families). Potential interviewees were asked for their individual informed consent. Village-level informants were sometimes keen to supply information, either because they felt the story of their suffering should be heard, or because their motives had been misunderstood. These informants often were willing to give their views in informal focus group settings. Other informants were willing to speak only anonymously and in private. Both kinds of interview were conducted in the Mende language, mainly by a single interviewer. Mende is a strongly oral culture where words are counted and remembered. Recording devices were used in village-level interviews sparingly or not at all. Notes were written-up as soon as possible after the interview.

Stories told were complex and multi-faceted. Several visits were made to key sites of enquiry, to cross-check material and hear other views. We also followed up events by interviewing responders. The Ministry of Health was notified about our study, and medical personnel generally granted requests for interviews. We were not able to get the agreement of security personnel deployed in quarantine operations at district level. In the accounts that follow we link events described to numerical data base sources where appropriate.

## Numerical data records

Numerical data relating to the West African epidemic of EVD derive from two main sources–admission data collected via (WHO standard) Ebola Case Investigation Forms, completed when a person suspected of having Ebola presented to a care or holding facility, and laboratory records of tests for Ebola applied to blood or swab samples.

The version of the standard case investigation form used in Sierra Leone had three pages. The first two pages are filled in on first contact or admission. There are sections for names, gender, address, occupation, location where the patient became ill, date on which they became sick, and symptoms and hospitalization. A second page covers contact with Ebola patients, including attendance at funerals, and outcomes (hospitalization and death). The third page provides formats for recording the results of laboratory tests.

Eye-witness reports confirm that medical personnel coping with the arrival of a patient with Ebola symptoms had limited opportunity to fill in or check the numerous details. Some patients were too distressed to supply necessary information and questions will often have been answered by family members or helpers, not always accurately, either because details were not known, or because information was deliberately concealed to avoid incrimination. Participation in funerals was declared illegal under Ebola emergency regulations and carried the risk of a fine or imprisonment.

That record keeping was often inconsistent can be illustrated by the case of Bo, the largest town in provincial Sierra Leone. Ebola cases arrived in Bo from Kenema in the early days of the epidemic. A rather large urban outbreak followed, eventually necessitating the opening of a dedicated Ebola Treatment Unit at the outlying village of Bandajuma. The records for Ebola victims in the Bo outbreak were supposed to indicate the district, chiefdom and settlement of origin. Bo town is the headquarters of Bo District and also of Kakua Chiefdom, one of Sierra Leone's 149 chiefdoms (the lowest level of government administration). The data base lists 181 laboratory-confirmed cases for Kakua chiefdom. Eight cases were wrongly assigned and came from other chiefdoms. Of the remaining 173 cases only 36 (38%) can be unambiguously assigned to Bo town from information in the data base. In 58 other cases (62%) a location in Bo town can be inferred only from the name of the street. Seemingly, there were 94 confirmed cases of EVD in Bo town, but this cannot be known except from fieldwork on the ground, with a street map in hand.

One further source needs to be explained, since it is not available elsewhere. This is a hand-written log of admissions, test results and outcomes maintained (apparently as a personal initiative) by a nurse at the Ebola Holding Centre (EHC) outside Moyamba town. The Moyamba EHC was hastily improvised facility arranged in an empty (school?) building with few affordances operating from July 2014. It handled many of the early cases of EVD in Moyamba District, until internationally staffed and supported Ebola Treatment Units (ETUs) were opened in Bo and Kenema in October 2014, at which point some Ebola+ cases were referred to the new ETUs. The EHC in Moyamba was replaced (in December 2014) by a better-equipped ETU staffed by international (Norwegian) volunteers, to which all remaining cases were transferred. A photocopy of the admissions log of the EHC from June to November 2014 was made

available to us after interviewing its compiler and proved useful in confirming details regarding dates and outcomes of cases referred to in village interviews. An anonymized version is included in the online supporting materials.

**Limitations of this study.** Any study such as this depends on the accuracy of informants' memories and their willingness to be frank. Events described were triangulated from accounts provided by independent witnesses, and discrepancies followed up and resolved. These witnesses were identified by the second author. They were independent in the sense that they were members of the community in question but did not belong to the immediate household of the infected person. In several communities we were also able to make use of base-line data collected by focus group and questionnaire interview for the Ebola Response Anthropology Platform during the epidemic in 2014 (publicly available online at www.ebola-anthropology. net). Again, inconsistencies were followed up and resolved. An inherent limitation of the approach is that it cannot infer larger regional trends. If the standard numerical approach had found evidence of larger regional trends this would have cast doubt on the utility of ethnographic analysis to understand heterogeneity through case study approaches like ours. Instead, however, numerical research to date has reported local heterogeneity, justifying the disaggregated case-following approach adopted here. Furthermore, we identify categories of elements that explain heterogeneity in this context, which are likely to have wider relevance in other contexts and can be tested in other settings.

## Results: Local infection chains analysis

Analyzed from the perspective of the standard numerical approach the various episodes making up the 2014–15 Ebola epidemic in Sierra Leone appear highly heterogenous. This heterogeneity can be seen in figures for outbreaks by locality (Table 1).

Western Urban (Freetown) and Western Rural (Freetown's peri-urban periphery), together accounting for about 30 per cent of the national population, had 41 per cent of all Ebola cases. Provincial Sierra Leone is divided into three provinces, 12 districts and 149 chiefdoms (the

**Table 1. Ebola in Sierra Leone by district, date of first case, average length of infection chains at chiefdom level, and total numbers of laboratory confirmed cases.**

| DISTRICT | DATE OF FIRST CASE | LENGTH OF INFECTION (days) | N OF CASES | |
|---|---|---|---|---|
| | onset of symptoms | | lab confirmed | |
| BO | 13/06/2014 | 90 | 315 | |
| BONTHE | 10/10/2014 | 32 | 6 | |
| BOMBALI | 06/07/2014 | 161 | 1049 | |
| KAILAHUN | 18/05/2014 | 133 | 524 | |
| KAMBIA | 12/09/2014 | 259 | 241 | |
| KENEMA | 13/06/2014 | 120 | 497 | |
| KOINADUGU | 29/08/2014 | 198 | 111 | |
| KONO | 27/06/2014 | 132 | 260 | |
| MOYAMBA | 02/07/2014 | 95 | 211 | |
| PORT LOKO | 02/07/2014 | 280 | 1202 | |
| PUJEHUN | 28/07/2014 | 67 | 31 | |
| TONKOLILI | 05/08/2014 | 151 | 489 | |
| Western Rural | 20/07/2015 | 300 | 1146 | |
| Western Urban | 25/06/2015 | 406 | 2274 | |
| **ALL PROVINCIAL** | | | 4936 | 59% |
| **ALL WESTERN AREA** | | | 3420 | 41% |
| **TOTAL** | | | 8356 | 100% |

lowest level of local government), and contained 59 per cent of cases, but these were unevenly distributed. According to the national data base, and including only laboratory confirmed entries, 39 chiefdoms (26%) had no cases at all. Of 110 chiefdoms with cases (74%) as few as 14 chiefdoms accounted for 60% of all chiefdom-level cases (5014 cases).

## Moyamba District infection chains: Families begin to keep their distance

S1A Village is a typical medium-sized off-road farming settlement on the left bank of the Taia river, in Kori chiefdom, Moyamba District. It is joined to the main Bo-Freetown highway by a 7 km track from the right bank of the river. The river must first be crossed by canoe. The track is motorable only in the dry season but can be used by motorcycle taxis (*okada*) at all times of the year. The people are Mende-speakers, but they live along the provincial boundary with Northern Sierra Leone and are intermarried with Temne-speaking families from the other side of the river. The Taia river floods in August and canoe traffic ceases for a period. The settlement is in effect cut off in the middle of the rains.

Ebola came to S1A Village as a spill-over from the Kenema outbreak. The virus crossed the border from Guinea in early 2014, and an outbreak in Kailahun District resulted in an Ebola case being brought to the Government Hospital in Kenema, where there was an isolation ward for victims of Lassa Fever. Ebola has even more exacting biosafety requirements than Lassa Fever, and the nursing staff in Kenema were not prepared to deal with Ebola in advance.

The WHO case definition of Ebola at that stage emphasized bleeding as a key sign. Few Ebola patients in Sierra Leone showed signs of bleeding, and cases were misdiagnosed as malaria or Lassa Fever. In addition, the authorities were slow to react to laboratory information that there was an outbreak of EVD in Sierra Leone and supplies of chlorine and personal protective equipment in the hospital were inadequate.

Kenema Government Hospital became the site of a major outbreak of nosocomial infection in June-July 2014. EVD appears to have come to S1A Village via a diamond miner (AA) working in Lower Bambara chiefdom, not far from Kenema. One of his sons (BB) fell sick and was taken to Kenema hospital, where he died. It is not known whether he was already sick with Ebola or became infected in the hospital.

His death was treated as Ebola and AA was denied sight of the son's body. The stunned father was taken ill at the hospital gates. His family had no wish to let him follow his son into Kenema hospital, perceived as the source of Ebola infection. They conceived a plan to seek medical treatment from a sister, who was a renowned herbalist in S1A Village in Moyamba District. AA was familiar with the village since it was his mother's place of birth and he himself came from a village nearby.

The government had tried to block the further spread of EVD by reinforcing the check point at the western entrance to Kenema, forcing passengers to submit to medical inspection intended to detect anyone with elevated temperature. Local transporters, however, are familiar with a number of by-pass routes opened during the war. One of them was tasked to evade the checks and deliver AA–described as profusely sweating—to a settlement in Kori chiefdom on the main road to Freetown, where a bike taxi was chartered to take the sick man–held fast as a pillion on the back of the bike by one of his sons—to the canoe crossing to S1A Village. There, too weak to climb up to the village, he was carried up the cliff path by a strong young volunteer, who confirmed details of AA's arrival in an interview in 2016. The date was reported as 9th July 2014, six weeks after the government had first announced the presence of the disease in the country.

People interviewed in S1A Village were doubtful whether AA had arrived with Ebola. Normally, the progression is a 3-day period of headaches and fever followed by a 3-day "wet"

period in which vomiting, diarrhoea and bleeding occur, before death or eventual recovery. Unusually, AA survived for two weeks after his arrival as a sick man, alternately sleeping in a family hut and in the mosque, while his sister treated him with various leaf infusions, before he eventually succumbed, and was buried just outside the village. He was not observed to be showing reported signs and symptoms of Ebola.

AA's sister (CC) then sickened. By mid-July the country was on high alert for Ebola cases. News of the sick woman had reached the Community Health Officer (CHO), head of the Ministry of Health team in Kori chiefdom, based in Taiama, and he arrived with a team on the right bank of the Taia river opposite the village, seeking a blood sample from the sick woman. The CHO phoned across to the chief (DD) explaining that he wanted to send two of his team across the river to collect the blood. The chief responded that this would not be possible as he had had no notification about the visitation, either from the Paramount Chief or the Ministry of Health.

The chief later explained, in interview, that he knew sick people were sometimes given blood, but he had never heard of blood being taken from the sick, especially on the point of death. Lack of explanation about blood testing, and the arrival of responders in full PPE, convinced some in the village that this mysterious new disease was a cover for a kind of medical "vampirism" (*bona hinda*).

The CHO was forced to withdraw but came back with his team and authorization the next day. DD required that the CHO cross the river to do the blood sampling in person. The sample was taken for testing, and proved positive, but the woman died before any follow up was made.

As a *sowei* (an elder of the women's Sande society) CC's burial attracted a group of her peers from surrounding villages. The burial was conducted according to the secret rites of the society. This was before the Ebola burial rules had been promulgated (August 8th 2014) and the CHO later explained in an interview that he did not have any means to prevent the funeral. Interviewees in S1A Village claimed that no further infections occurred among the group of women conducting the burial rites.

There were, however, further cases of Ebola both in S1A Village and in S1B Village, a small settlement on the left bank of the Taia, about 3 km to the south, and at least two of the corpses were swabbed on burial. In all, village people report that there were 22 deaths subsequent to the case of CC in S1A Village (14 female, 8 males) and nine deaths in S1B Village (7 females, 2 males, with six persons coming from one family related to the family of CC in S1A Village). Several of the people infected in S1B Village had visited CC to express sympathy with her, and later to take part in her funeral. These visits are an inescapable social obligation in tightly inter-married rural communities.

The evidence that these deaths were Ebola cases is not conclusive but is supported by the Moyamba records. We were able to cross-reference five persons who were admitted to the facility on 24 September 2014 bearing the same family name as several of the persons who died in S1B Village. They are listed as coming from Taiama, the chiefdom HQ for Kori chiefdom; S1A and S1B Village are inaccessible places, and Taiama is the nearest settlement of any size. Of this group of five patients, two were diagnosed as positive for Ebola, two died without diagnosis, and one (an old man) was negative, and later interviewed for this study.

A youth organizer in Taiama (EE) reported that he had helped to arrange the transport of two suspected Ebola cases from the right bank of the Taia opposite S1B Village. Even if an Ebola ambulance had been available it would not have been able to travel along the track to the riverbank in rainy season conditions, so a commercial motorbike rider had been hired instead. Motorbikes crossed the difficult places on single track bridges improvised by the commercial riders. The incident was vivid in his mind, because the rescue team arranging to collect one of the patients had only a single PPE suit, and the bike rider and the pillion passenger needed to

support the patient were both nervous of becoming infected. A nervous rider on a difficulty rainy season track was more likely to have a spill, with serious consequences for the patient and everyone else in the team, so the decision to dress the patient in the PPE, rather than decide which of the two–rider or supporter–should wear the protective suit, made sense. EE told the story self-deprecatingly, remarking 'how little we knew about the disease at the time'. In fact, it suggests that everyone engaged in extracting high-risk patients was beginning to understand about the significance of body contact in spreading the disease.

An elderly man interviewed in S1B Village (FF) told us he had been admitted to the Moyamba holding centre along with two other persons from his village, both of whom he knew had tested positive for Ebola and subsequently died. FF was tested Ebola negative and had come back to the village to give notice of the fate of the two persons with whom he had been admitted. Clearly, some of the deaths were confirmed as EVD through laboratory evidence. But still villagers have doubts about whether all admissions had Ebola on entry, or were cross-infected after arrival, due to the notoriously poor conditions at the Moyamba facility.

Quarantine was imposed in S1A Village when a small detachment of five police and army personnel arrived on 18th August 2014 and stayed for a month. The recently enacted national emergency regulations authorized military intervention in epidemic response. If the admission of a group of patients "from Taiama" to Moyamba EHC on 24th September 2014 corroborates our interviewee's recollection in S1B Village then infections must still have been occurring in S1A and S1B Village in the second half of September. So why the security forces were withdrawn about that time is unclear. The national data base records dates for onset of symptoms for patients with positive blood tests in Kori chiefdom from 2nd July to 24th November 2014, with two outliers in January and February 2015. One person from S1A Village and five from S1B Village are listed with positive blood samples.

What is clear from interviews is that local perceptions were changing very quickly. There was now a distinct awareness the disease was spread by body contact. The name for EVD given in S1A Village was *bondawote* (literally "family turn away"), glossed by an informant as meaning "you are completely abandoned to die". Another interviewee said that "people ran to their farms, [and] most of those who got sick recover[ed] when they stopped touching each other". Youth leaders in a village at the end of a river-bank track leading to S1A and S1B Village feared that sick relatives in the two afflicted villages would seek help from the health post in their village. They went out and cut down the stick bridge crossing a rainy season flooded ravine, making further contact impossible. This rendered apparent in dramatic terms the rapidly acquired notion that "touching" was deadly. It is likely that this local "turning away" was a significant factor in ending local infection chains.

## Spread of infection to a local market center

S1C Village is a small settlement on the main Bo-Freetown highway in Kori chiefdom at the point where the c. 10 km track branches leading to the right bank of the Taia river opposite S1A and S1B Village. This was where AA the sick man from Kenema, transferred from a taxi to a motorbike on his journey to reach his relative, the herbalist, in S1A Village. AA was uncle to GG, a woman living in S1C Village, who joined the CHO's the team on the trip to collect the blood sample in S1A Village. It is reported that she went to S1A Village to warn her relatives there about the dangers from Ebola. How she became infected is not clear. She tested positive for EVD in Taiama and died there on 12th September 2014. Her sick husband sought treatment from a pharmacist based at road junction market (S1D Village) on the highway to Freetown (Fakuniya chiefdom). The pharmacist also sickened and died, presumably from Ebola, but without test data to confirm it. The man was a member of a Catholic sodality [closed

association] and his funeral attracted sympathizers from as far as Freetown and Bo. This was followed by a substantial outbreak of EVD in S1D Village, with spread of cases up and down the Freetown and Moyamba roads, perhaps reflecting local networking among traders. The first 12 patients testing positive for Ebola from S1D Village were admitted to Moyamba EHC on 17[th] September. In all, 36 patients admitted to Moyamba EHC from S1D Village tested positive for Ebola. S1D Village was quarantined by the security forces. The last E+ case admitted to Moyamba EHC is recorded on 8[th] November 2014. The national data base shows 64 E + cases from S1D Village and 11 from adjacent villages (some in Kori chiefdom). The dates for onset of symptoms of the first and last cases in Fakuniya chiefdom are 9[th] September 2014 and 4[th] November 2014.

## Bo District infection chains: Good response marred by accidents

S2A Village is a village about one km. south of the Kenema-to-Bo highway, some 20 km. east of Bo city. It is a centre of Islamic instruction. A noted teacher (*kamoh*) from a village on the border of Bo and Kenema districts was offering instruction in S2A Village and then fell sick. Enquiries in his home village suggested he had been in the habit of going to Kenema for treatment for some longer-term medical complaint. It can be surmised he became infected with Ebola as a result of the outbreak in Kenema town.

The *kamoh* died in S2A Village on 13[th] August and was buried according to Islamic practice. This requires thorough washing of the corpse. The implications of the brand-new Ebola national regulations had yet to be realized in the village, even if they were known. Some of his pupils are said to have used water from the washing of their learned master's body in the hope of inheriting some of his wisdom and charisma.

The *kamoh* had been seen by medical personnel in a near-by health centre but he had been discharged without any diagnosis of Ebola or instructions about what to do if is condition worsened. Soon after the burial his pregnant wife (HH) began to complain of joint pains and fever. She turned for help to the mother-and-baby unit in near-by S2B Village. The midwife in charge examined her and offered some treatment for the fever. She had already received a briefing about Ebola risks, and used gloves in her examination. But HH's symptoms were as yet no different from malaria, so she was discharged and sent home with appropriate medicine for her presumed condition.

Two days later HH died, after giving birth to a still-born child. Her other two children sickened and died two days later. The woman's death was reported to the Paramount Chief for Kakua chiefdom, based in Bo, and the medical authorities immediately intervened. Eleven more people fell sick in rapid succession. Samples were taken and inter-village movements were stopped. The villagers were told that even the nurses who had treated HH would now have to be quarantined for 21 days, and that burials could only be undertaken by a specially equipped burial team.

A brother of the *kamoh* died and his body was not buried for 3 days, something that was especially shocking to the villager's Islamic religious sensitivities. The hazard suits of the burial team alarmed villagers, as did the unexpected arrival of police, military and the District Medical Officer.

By now a further 13 people had become seriously ill and were taken to Bo Government hospital. There were no preparations to deal with Ebola cases, and the sick villagers were placed in a kind of holding shed. There was no infection of hospital staff or patients, but the episode was disturbing to community members.

One remarked that "*to my dismay no treatment was given but [they were] just cluster[ed] into a non-caring room, where five of them immediately died*". They were told that there was no

bed for admission at the hospital. The district Ebola task force then decided that the sick villagers should be brought back to the village, to be quarantined in the community school.

The quarantine was strict. The security forces prevented any movement, even for the essentials of daily life. Families suffered heartbreak as they heard their loved ones crying out for water and were prevented from helping. Nor was there any satisfactory arrangement to feed the patients. Villagers feared the security forces had been ordered to poison them.

The situation was improved when the local parliamentarian arranged for the delivery of beds, and (after 12 days) "*9 bags of rice were given to the community with the population of 210, including children*". The district Ebola task force helped to mobilize supplies, even though at this stage it lacked a budget and remained reliant on voluntary contributions.

New cases were still occurring, and the death toll continued to rise. Villagers formed three burial teams, so that they could bury their own loved ones promptly, and the authorities seem to have concurred in this. Thirteen victims were taken to holding centers in Kenema and Kailahun, and only one person is said to have survived.

The total number of laboratory-confirmed cases of Ebola in S2A Village appears, from the national records, to have been 43 (over 20 per cent of the total village population, and 28 per cent of our corrected number for all laboratory-confirmed Ebola cases in Kakua chiefdom, including Bo city). The villagers reported in interviews that there were 37 deaths. This implies either that only six people survived or that the national data base under-estimates the total number of infections in S2A Village. In fact, the names of seven certificated survivors were reported during our enquiries, suggesting the discrepancy may not be large.

The strict quarantine was not wholly effective. There was at least one escape, when a young man (II) broke bounds to visit a larger settlement, T, a small town about 12 km. south of Bo. II then showed symptoms of EVD. Whether he was infected in S2A Village or on his arrival in T is unclear.

T is close to Bo city where cases are first recorded in the national data base from mid-June. The national data base records 29 laboratory-confirmed cases in T over the period 23rd July to 13th December. The outbreak here is probably connected to infection in Bo. We have no date for the quarantine breach in S2A Village but it is unlikely to have been earlier than September since quarantine was imposed from the second half of August.

T was a bridge to a somewhat larger outbreak in the adjacent chiefdom, Bumpeh Ngao, where 48 laboratory-confirmed cases are reported in the national data base. The Bumpeh outbreak occurred at a later stage in the epidemic, when communities and responders were better prepared. The infection chain ran from 3rd October 2014 to 10th January 2015, a period of 100 days. The Community Health Officer thought that most cases had some connection with a large funeral for a "big person" (Mende: *numu wa*) in T.

At S2C Village in Bumpeh Ngao chiefdom there is a long-established and well-respected mission hospital. The hospital authorities approached the local community and explained that the hospital (which lacked a ward capable of handling Ebola cases, until international responders built one) would have to close if cases arrived.

It was agreed with the chiefdom authorities that in order to keep the hospital open during the Ebola outbreak all potential patients would pass through a screening and triage process located outside the hospital. Any potential patients showing signs and symptoms of Ebola would be conveyed to newly opened Ebola case-handling facilities in Bo and Kenema, where they would receive specialist treatment. Chiefs and sub-chiefs were responsible for conveying a message about why these measures were necessary and the message was widely understood.

Not all families agreed with the implications, however. JJ, the female chief of a satellite settlement (S2D Village), attended the "big person's" funeral in Village T, where she appears to have contracted Ebola. The nurse in charge of the village health center suspected the true cause

of her illness and informed the authorities, but JJ's family objected. Nevertheless, an ambulance was called, and she was taken to the case-handling facility at Bandajuma, where she died. Disagreement over responsibility for JJ's fate led to a breakdown in relations between the family and the nurse so severe that she had to be transferred to another district.

JJ was not alone in trying to hide her symptoms. Interviews with medical personnel in S2C Village and an adjacent community health post elicited several other stories about the lengths to which other patients went to hide symptoms to avoid being transferred to an Ebola case-handling facility. As yet, there were few survivors from such centers, which were suspected of being "death camps".

Informants reported that international advisers wondered why Bumpeh Ngao chiefdom was "difficult". The fear of being wrongly diagnosed as an Ebola case and cross-infected in a holding or treatment facility grew as stories about Kenema hospital and poor conditions at the holding centre in Moyamba spread. Local doubts over diagnosis were reinforced by a case in which a blood sample sent by the hospital in S2C Village to a laboratory in Kenema had come back wrongly categorized as negative because it had been confused with a sample from a village with the same name in Kenema District.

## Null infection chains: A dog that didn't bark?

Thirty-nine chiefdoms had no cases of Ebola; many were protected from spread of infection by distance or poor roads. Four chiefdoms, however, were situated on or close to the main national road transport network, surrounded by chiefdoms with cases. One of these is Kamajei chiefdom in Moyamba district. Led by their Paramount Chief community activists in Kamajei closed tracks and closely monitored movements of strangers. Interviews and surveys undertaken in this chiefdom during the epidemic (fieldwork in December 2014, www.ebola-anthropology.net) showed widespread acceptance that Ebola was spread by body contact, and not through consumption of bush meat as people had earlier been informed.

It is also relevant is to ask about communities where cases occurred, but where infection chains were closed down promptly. One such chiefdom is Niawa Lenga in Bo District. This was one of four chiefdoms (in Bo and Moyamba Districts) in which an Ebola infection chain lasted for less than 50 days. The other three chiefdoms in this group (Bagbo, Bagruwa and Jaiama Bongor) had few confirmed cases, but Niawa Lenga had a substantial number (19).

In general, chiefdoms in Bo and Moyamba districts were among the quickest in the country to end infection chains, with an average per chiefdom of 90 days for Bo district, and 95 days for Moyamba, compared to figures for chiefdoms in the first and last districts to experience the outbreak–Kailahun and Port Loko—with an average of 133 and 280 days respectively.

Why was the outbreak in Niawa Lenga ended promptly? The answer appears to be that by mid-October 2014 the Standard Operating Procedures (SOPs) for Ebola control of the Bo District Ebola Response team were fully implemented and working successfully, and that there was little or no prior opposition to and distrust of local authorities, a factor in a case described in detail by Parker et al. [9]. Lessons from S2A Village had been well-learnt.

Fieldwork revealed two distinct infection chains. One involved the chiefdom headquarters, S3A Village and the other a village, S3B Village which is close to the motor road from Bo to Yele. Vehicles to S3A Village take the motor road, and branch right just after the town of Dambara. There is a shorter route–a track leading to S3A Village from the northern outskirts of Bo.

A middle-aged female resident of Bo (KK) was heading along this track, possibly seeking local treatment in S3C Village for a long-standing complaint. S3C Village is located on the boundary between Kakua and Niawa Lenga chiefdoms. Here, KK was taken ill with what turned out to be EVD. Her helpers decided to hire some youths to carry her in a hammock to

her family. The hammock party left for S3A Village before dawn, without informing the town authorities, a breach of local protocol suggesting they had something to hide.

Discharging her hammock carriers at the entrance to S3A Village and being too weak to walk, KK sent for family helpers to bring her to her house, where she died a day or two later. Others in the home were infected, including a child who had slept on KK's bed. The woman's death was promptly reported to the district response team and the death was confirmed as Ebola; SOPs were promptly activated.

The community was quarantined–a process supervised by a doctor from the international response. At first security forces kept out all visitors, including relief workers, but villagers complained to the foreign doctor and food and other necessities were quickly supplied.

Niawa Lenga is a chiefdom of small-scale rice farmers, and October is the harvest period, when daily life centers around the rice farm. Some people doubtless slipped away down unregulated bush tracks and made themselves quietly absent in their farms, while others collected daily necessities from the relief agencies. The demand by the chiefdom authorities for adherence to Ebola byelaws was respected, and no disputes were reported. Infection in S3A Village ceased.

Infection in S3B Village appears not to be connected with the outbreak in S3A Village. The origins of this second outbreak lie in Bo Government Hospital. A nurse contracted Ebola (apparently outside the hospital) and patients on her ward took flight. One of the patients, LL, headed home to S3B Village. A cluster of cases subsequently occurred in S3B Village and control measures were rapidly implemented by the Bo-based District Ebola Response team, which included establishing quarantine barriers. With a local politician's help quarantined homes were supplied, enabling people to stay put.

There was one hiccup. At one of the burials the "safe burial" team attended without a stretcher and improvised with sticks from a farm. The farmer collecting these sticks after the burial became another victim of the outbreak.

The village location records show 12 cases in Niawa Lenga chiefdom, 4 in S3A Village and 8 in S3B Village. The records of laboratory confirmed cases show 19 cases in Niawa Lenga between 17[th] October and 21[st] November 2014. Villagers in S3B Village insist there were more, with 11 deaths in their village alone. Nevertheless, excepting for the mistake over the stretcher, control measures worked as intended. The first confirmed case was recorded on 17[th] October 2014 and the last on 21[st] November. The outbreak in the chiefdom was controlled within 36 days.

## Discussion

The results reported here help confirm a picture of the Ebola epidemic in Sierra Leone as a concatenation of smaller outbreaks. The heterogeneity of the epidemic thus needs to be explained in terms of local behavioral circumstances and, most especially, social circumstances [10], for which a standardized top-down numerical analysis or externally driven emergency-response is not necessarily the most effective [11]. This was certainly true for our two study districts, which were affected early in the epidemic and had to cope before a national or international response had been fully mobilized.

The local name for Ebola (in the Mende language of the south and east of Sierra Leone) is *bondawote*–"family turn around". This recognizes an essential truth about Ebola infection; it is a disease of social intimacy, with close family members bearing the highest risks of further infection [7]. The risks of infection peak in the final "wet" phase of the disease and immediately after death, when the corpse is prepared by family members for interment.

The disease also makes inter-community jumps [8], but our ethnographic data reveal that these jumps are often the result of family networking. Control, or loss of control, over infection

thus depended a great deal on cooperation of families, and in particular on whether family care givers were persuaded to collaborate in reducing risks of contact with a sick person or infected body.

The disease came under control only when the family "turned around". The heterogeneities of Ebola infection dynamics reflect whether families turned quickly or not, and whether this "turning" was willing or achieved only under pressure. Four different groups of explanatory elements can be discerned in the data presented. As overarching themes emerging from our detailed analysis, they help explain local heterogeneity in ways that need to be considered for other localized outbreaks (either in other parts of Sierra Leone, or in other countries and other epidemics).

## 1. Heterogeneities caused by externalities

Whether an epidemic chain was controlled or not sometimes depended on externalities–events over which there was little or no control. A good instance of this would be the isolated location of S1A Village and the seasonal flood of the Taia river, which inhibited initial response and delayed diagnosis.

Diagnostic accidents–the mix up of blood samples from two different villages with the same name–also help explain why some outbreaks were larger than might have been expected from the current state of knowledge or preparedness on the part of responders. The surge in cases in Bumpeh Ngao, quite late in the epidemic's trajectory, when response modalities were better developed, is a case in point. Misdiagnosis created a false confidence among family carers that a diagnosis of Ebola was unlikely.

Panic over nosocomial infection might also be considered a kind of accident, connected to the totally unexpected outbreak of a disease never before seen in the region. The spread of Ebola to patients in the isolation ward at Kenema Hospital, and the notoriously poor conditions in the makeshift Ebola Holding Centre in Moyamba, then led to a sudden and widespread collapse of confidence in medical treatment. Some people became convinced that Ebola was spread deliberately by medical workers, others that Ebola case handling was connected to theft of body parts and blood. Families strategized desperately to prevent patients being consigned to Ebola handling facilities.

Much of the Ebola response was a race against time to put in place proper procedures, in which improvisation was often required. Allocating the single PPE suit to the patient, rather than to rider and pillion helper when extracting patients from S1B Village, might be seen as an inspired solution. The burial team's lack of a stretcher–did they forget to pack it in their haste? —in S3B Village was a mistake with which a village farmer paid with his life.

## 2. Heterogeneities resulting from structural or health system deficiencies

Families often struggled to bring sick patients for diagnosis, and many helpers were at risk of infection during that process. In some instances, little could be done to improve access to services (S1A and S1B Village, for example). Later in the epidemic, better transport equipment (dedicated Ebola ambulances with trained crew, for example), improved communication (notably, a telephone helpline), and more rapid, mobile laboratory diagnosis certainly improved capacity to reach and transfer patients over a large part of the country.

The availability of Ebola case handling capacity, and the degree to which communities trusted or shunned those facilities [12], varied locally, and negative impacts are clearly seen in our data. A study of the Kenema Ebola Treatment Unit (open from October 2014) reports that it only gained wider acceptance after the first survivors returned to their families [13]. The facility at Bandajuma (Bo) also opened in October 2014, and its presence and impact are glimpsed only in our later case studies. The Ebola Holding Centre in Moyamba town was

opened as early as June 2014 and left its (negative) mark across the district. It was eventually replaced by a Norwegian-staffed and funded ETU in Moyamba in December 2014, too late for the infection chains reported above.

The case-study material relating to S1A and S1B and S2A Villages illustrate the disadvantages of not yet having in place a clear set of Ebola response SOPs, and a proper relief system supply chain to support quarantine. The benefits of having such organizational procedures are apparent in the speed with which the later Niawa Lenga outbreak was contained.

The original case definition for Ebola imported from central Africa over-emphasized bleeding as a sign and symptom. The case definition was later changed in line with actual experience, but families varied in the extent to which they took account of this earlier misinformation. Arguments about symptoms emerged where precipitate action in treating a potential case as Ebola was resisted (we documented this in S3C Village, for example).

Other examples of poor messaging, inappropriate in the local context, exist. For example, bush meat, especially eating of monkeys, was widely warned against over radio and on posters as a cause of infection. Villagers who never ate bush meat (often for religious reasons) imagined themselves to be safe from infection. It was only late in the epidemic that this message was replaced by an emphasis on limiting body contact with persons of unknown Ebola status. Village communities varied in the extent to which they worked out for themselves whether or not bush meat or body contact were risks (www.ebola-anthropology.net).

## 3. Heterogeneities linked to cultural considerations

Older family members are often heavily involved in caring for and treating the sick and advising on steps to be taken in case of serious sickness and death. This explains the quantitative finding that older people were disproportionately at risk of infection from EVD. There was then a knock-on effect–older people in positions of family leadership are more likely to be senior members of the major male and female sodalities. The elaborate funeral rituals of the sodalities played a significant part in local multiplication of infection (e.g. S1A and S2A Villages).

It is not the size of the funeral that determines the infection risk, but the distribution of duties in preparing and taking leave of the corpse [7]. These are matters known only to members of the sodalities. The pattern of subsequent infections might then depend on where the key elders came from across a chiefdom or chiefdom section. The conversation between communities and responders about control of infection risks from funerals improved only when sodality members with information on Ebola infection control talked to sodality elders, who then turned this information into safer practical outcomes [7, 9].

Local government in rural Sierra Leone is dualistic. Chiefdom law is separate from national law. National government intervenes locally only with the collaboration of chiefdom administrations. A slip-up over sending notifications to the local chief through the correct channels hindered the process of establishing that there was EVD in S1A Village and this delayed implementation of infection control.

Belief in the efficacy of traditional herbal medicine is high in rural areas of Sierra Leone, and this trust was intensified by experience of hospital-based nosocomial infection. Traditional practitioners vary in reputation. A renowned practitioner will draw clients from far and wide. The likelihood of a practitioner being infected by a patient is greater where the catchment is wide. Ebola first spread across the border from Guinea through the patients of a well-known herbalist. The index case in S1A Village was also a renowned herbalist. Villages without noted practitioners were less likely to experience infection. The government banned traditional medical practice for the duration of the epidemic. Arguably, a better approach might have been to find a role for herbalists in the epidemic, perhaps as community interlocutors, to explain the infection risk.

## 4. Heterogeneities as a result of variations in coping options

Once communities realized that Ebola infection risks were linked to contact with infected persons, they began to develop ideas for limiting social contacts, especially with strangers. One approach was to impose self-isolation by controlling entrance to and exit from the community, building on experience with civil-defence during the civil war (1991–2002). Another approach (again widely practiced during the war) was to retreat into *sokoihun* ("corners"). Typically, this would mean the household withdrawing to the hut in the family rice farm, where there was both food and shelter. These solutions were less easily applied in some places than others. The greater the dependence on trade rather than the farm for subsistence livelihood the less practical it was to practice self-isolation.

Community members in S1A Village explicitly mentioned the self-isolation option, more or less reinforced by the cutting of the footpath bridge linking them to S1C Village. Informants connected living in "corners" with cutting down on bodily contact. S2A Village, however, was more dependent on its external links, as a center for both trade and Koranic education, and had no option but to accept externally enforced quarantine, and to apply for relief assistance. More generally it seems clear that the greater difficulty in ending infection chains in districts closer to Freetown (notably Port Loko and Western Rural) relates to the greater involvement of these districts in trade and transportation.

## Conclusion

The overall conclusion is that the epidemic of Ebola Virus Disease in Sierra Leone 2014–15 is best viewed from a disaggregated perspective, as a series of local, but linked episodes, shaped by a diverse series of factors including bad luck and miscalculation, as well as variation in local cultural imperatives, response strategies and configurations of local livelihood opportunities. Despite local specificity, we have identified four groups of factors that help explain this heterogeneity which are likely to have relevance for other settings. It is critical that epidemic-response rapidly captures the presence of such heterogeneity. To do this implies the continuing need for an ethnographically informed epidemiology: ethnographic research is critical for developing a richer picture of epidemic spread and a better understanding of epidemiological data. Integrating these disciplines in research is challenging, but these are important epistemological considerations that extend well beyond the outbreak of Ebola in West Africa. Critically, understanding this heterogeneity will enable nuanced responses to outbreaks likely to be more effective and better received by the population.

Indeed, given current debates around evidence in the COVID-19 pandemic, we offer important reflections on how, in collaborations between epidemiologists and social sciences, the complementarity in collection, interpretation and use of qualitative and quantitative data, can produce not only more context-appropriate responses but also more accurate efforts to explain and model an epidemic taking into account social dynamics. Our findings raise questions about what evidence and whose knowledge "counts" [14, 15]. A key implication of our analysis is the necessity for local knowledge and inputs to be incorporated in the planning of any future outbreak responses–including in the current Covid-19 outbreak. This would involve working closely with key local figures coming from, or based in, the study communities in order to obtain a full picture of what has happened, why, and how outbreak responses can be made more compatible with local realities. It is also important that local nuance informs subnational and national response planning, with local data aggregated at subnational level to identify geographical patterns of disease. Of immediate relevance, in terms of future preparedness and current Covid-19 response, is the training of local researchers, with detailed

knowledge of local cultural contexts, in ethnographic field methods, to join teams carrying out real-time epidemiological analysis and help re-frame research as part of crisis response efforts.

## Supporting information

**S1 Village.**
(DOCX)

**S2 Village.**
(DOCX)

**S3 Village.**
(DOCX)

**S1 Data.**
(XLSX)

**S2 Data.**
(XLSX)

**S3 Data.**
(XLSX)

## Acknowledgments

Tommy M. Hanson is thanked for helping to carry out interviews in Bumpeh Ngao chiefdom in 2017. Foday Mamoud Kamara and Esther Yei Mokuwa are thanked for helping to carry out interviews in Niawa Lenga chiefdom in 2018. Dina Balabanova, Melissa Parker and two referees for the journal are thanked for their helpful comments on the manuscript.

Members of the Ebola Gbalo research team are Lawrence S Babawo, Dina Balabanova, Johanna Hanefeld, Tommy M Hanson, Baigeh Johnson, Foday Mahmoud Kamara, Bashiru Koroma, Susannah H Mayhew (PI), Alfred Mokuwa, Esther Yei-Mokuwa, Melissa Parker, Paul Richards, Ahmed Vandi.

## Author Contributions

**Conceptualization:** Paul Richards, Gelejimah Alfred Mokuwa, Ahmed Vandi, Susannah Harding Mayhew.

**Data curation:** Susannah Harding Mayhew.

**Formal analysis:** Paul Richards, Gelejimah Alfred Mokuwa, Ahmed Vandi, Susannah Harding Mayhew.

**Funding acquisition:** Susannah Harding Mayhew.

**Investigation:** Paul Richards, Gelejimah Alfred Mokuwa, Ahmed Vandi, Susannah Harding Mayhew.

**Methodology:** Paul Richards, Gelejimah Alfred Mokuwa, Ahmed Vandi, Susannah Harding Mayhew.

**Project administration:** Susannah Harding Mayhew.

**Resources:** Susannah Harding Mayhew.

**Software:** Susannah Harding Mayhew.

**Supervision:** Paul Richards, Susannah Harding Mayhew.

**Validation:** Paul Richards, Gelejimah Alfred Mokuwa, Ahmed Vandi, Susannah Harding Mayhew.

**Visualization:** Paul Richards.

**Writing – original draft:** Paul Richards, Susannah Harding Mayhew.

**Writing – review & editing:** Paul Richards, Susannah Harding Mayhew.

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
