## [Decision Letter · Decision Letter 0]

7 Apr 2020

PONE-D-20-06525

Re-analysing Ebola spread in Sierra Leone: the importance of local social dynamics

PLOS ONE

Dear Paul,

Thank you for submitting your manuscript to PLOS ONE. After careful consideration, we feel that it has merit but does not fully meet PLOS ONE’s publication criteria as it currently stands. Therefore, we invite you to submit a revised version of the manuscript that addresses the points raised during the review process.

We would appreciate receiving your revised manuscript by May 22 2020 11:59PM. To enhance the reproducibility of your results, we recommend that if applicable you deposit your laboratory protocols in protocols.io, where a protocol can be assigned its own identifier (DOI) such that it can be cited independently in the future. For instructions see: http://journals.plos.org/plosone/s/submission-guidelines#loc-laboratory-protocols

We look forward to receiving your revised manuscript.

Kind regards,

Mary Hamer Hodges

Academic Editor

PLOS ONE

Journal Requirements:

2. Please include your tables as part of your main manuscript and remove the individual files. Please note that supplementary tables (should remain/ be uploaded) as separate "supporting information" files

4. Your ethics statement must appear in the Methods section of your manuscript. If your ethics statement is written in any section besides the Methods, please move it to the Methods section and delete it from any other section. Please also ensure that your ethics statement is included in your manuscript, as the ethics section of your online submission will not be published alongside your manuscript.

Reviewers' comments:

Reviewer's Responses to Questions

**Comments to the Author**

1. Is the manuscript technically sound, and do the data support the conclusions?

Reviewer #1: Partly

Reviewer #2: Yes

2. Has the statistical analysis been performed appropriately and rigorously? 

Reviewer #1: N/A

Reviewer #2: Yes

3. Have the authors made all data underlying the findings in their manuscript fully available?

Reviewer #1: Yes

Reviewer #2: No

4. Is the manuscript presented in an intelligible fashion and written in standard English?

Reviewer #1: Yes

Reviewer #2: Yes

5. Review Comments to the Author

Reviewer #1: The authors present an ethnographic study of the transmission of Ebola Virus Disease (EVD) in two districts of Sierra Leone between June – December 2014 with the objective of enhancing understanding of the heterogeneities of disease transmission observed in the extant quantitative epidemiologic data. The study methods and rationale are, in general, adequately described, however a section of the “Limitations of this study” (lines 270-275) is unclear.

The narrative of the Results section should more clearly and concisely demonstrate linkages in the chains of transmission. This section would be enhanced by a map to demonstrate the geographic relationship of villages to each other, and epidemic curve of cases in the two to provide better context for the ethnographic descriptions. Table 1 is not labelled as such, and Figure 1 could benefit by having all data points labelled. Description of transmission of EVD in Village 3C (lines 628-640) should be enhanced so as to explain the relevant rapid disease control in this location.

The Discussion and Conclusion are generally sound; however, the authors could make clearer linkages conclusions and the supporting data. The initial section of 2. Heterogeneities resulting from structural or organizational deficiencies (lines 696-707) is unclear and should be re-worked. The overall conclusion (lines 798-807) is well laid out, but given its importance, the authors should elaborate upon their recommendations for the use of ethnographic methods is future outbreaks.

Reviewer #2: This is an important article, that makes a significant contribution to our understanding of the West African Ebola outbreak by reconstructing infection chains through rich ethnographic data to help explain local heterogeneity. The article however has the potential to state its contribution even further, and I would encourage the authors to do so more explicitly. As the authors already hint at, they are elucidating the importance of ethnographic perspectives in developing a richer picture and a better understanding of epidemiological data. This has important epistemological implications that extend well beyond the outbreak of Ebola in West Africa.

Indeed, given current debates around evidence in the COVID-19 pandemic, the article offers important reflections on how collaborations between epidemiologists and anthropologists, the combination of qualitative and quantitative data, can produce not only more context-appropriate responses but also more accurate efforts to explain and model an epidemic taking into account social dynamics. What becomes visible through the stories collected seems to me to speak to a larger point about what evidence and whose knowledge counts (and which *should* count) in the ways we come to know epidemics.

Of course the article rightly argues against a one size fits all approach but it would seem to me that the conclusion is not simply 'context matters' or 'every case is different', there are clearly, as the authors state, a number of 'categories of factors' that can help explain local heterogeneity in ways that need to be considered for different localised outbreaks (either in other parts of Sierra Leone, or in other countries and other epidemics).

The broader implications of this approach are already in the text, but in order to state their full potential it might be useful to include a more explicit consideration integrated right from the introduction and then more streamlined through the discussion and conclusion, rather than just a sentence at the end. A theoretical reflection on the 'social life of data' and the challenges of integrating this kind of interdisciplinary collaboration and local knowledge into an outbreak response may be beyond the scope of the paper, but could be hinted at as scope for future reflection.

Some smaller points more specific to the text:

- The data is acknowledged to be from the beginning of the response, and there are some interesting reflections on how it might have been affected by different kinds of intervention. Because the data can't speak to this, it would seem premature to suggest that externally driven emergency response is not the most effective (line 647). It would of course be fascinating to have the same level of depth of data across districts and from later on in the response, and this may be possible for a future paper.

- A brief note on ethical consideration more directly on the handling of patient data on hand-written logs, test results and the private records of a nurse in Moyamba, would be important

- Does Fogbo (line 384) need to be anonymised?

6. PLOS authors have the option to publish the peer review history of their article (what does this mean?). If published, this will include your full peer review and any attached files.

Reviewer #1: No

Reviewer #2: No

---

## [Author Response · Author response to Decision Letter 0]

14 May 2020

The letter has been checked. But we have changed our data availability statement. Anonymised sets of field notes will be held in the institutional data repositories of the London School of Hygiene and Tropical Medicine, and in the institutional data repository of the National Ebola Museum and Archive at Njala University Sierra Leone. URLs and doi will be supplied after acceptance of the paper for publication.

---

## [Decision Letter · Decision Letter 1]

3 Jun 2020

Re-analysing Ebola spread in Sierra Leone: the importance of local social dynamics

PONE-D-20-06525R1

Dear Dr. Richard,

We’re pleased to inform you that your manuscript has been judged scientifically suitable for publication and will be formally accepted for publication once it meets all outstanding technical requirements.

Kind regards,

Mary Hamer Hodges

Academic Editor

PLOS ONE

Additional Editor Comments (optional):

Reviewers' comments:

Reviewer's Responses to Questions

**Comments to the Author**

1. If the authors have adequately addressed your comments raised in a previous round of review and you feel that this manuscript is now acceptable for publication, you may indicate that here to bypass the “Comments to the Author” section, enter your conflict of interest statement in the “Confidential to Editor” section, and submit your "Accept" recommendation.

Reviewer #1: All comments have been addressed

Reviewer #2: All comments have been addressed

2. Is the manuscript technically sound, and do the data support the conclusions?

Reviewer #1: (No Response)

Reviewer #2: Yes

3. Has the statistical analysis been performed appropriately and rigorously? 

Reviewer #1: (No Response)

Reviewer #2: N/A

4. Have the authors made all data underlying the findings in their manuscript fully available?

Reviewer #1: (No Response)

Reviewer #2: No

5. Is the manuscript presented in an intelligible fashion and written in standard English?

Reviewer #1: (No Response)

Reviewer #2: Yes

6. Review Comments to the Author

Reviewer #1: The manuscript has been revised to adequately address previous concerns. It is suitable for publication.

Reviewer #2: I am satisfied that the revisions address my initial comments and I would recommend the paper is accepted

7. PLOS authors have the option to publish the peer review history of their article (what does this mean?). If published, this will include your full peer review and any attached files.

Reviewer #1: No

Reviewer #2: No

---

## [Editor Report · Acceptance letter]

26 Oct 2020

PONE-D-20-06525R1 

Re-analysing Ebola spread in Sierra Leone: the importance of local social dynamics 

Dear Dr. Richards:

I'm pleased to inform you that your manuscript has been deemed suitable for publication in PLOS ONE. Congratulations! Your manuscript is now with our production department. 

Kind regards, 

on behalf of

Dr. Mary Hamer Hodges 

Academic Editor

PLOS ONE